# Interaction of Impulsivity, Attention, and Intelligence in Early Adolescents Born Preterm without Sequelae

**DOI:** 10.3390/ijerph18179043

**Published:** 2021-08-27

**Authors:** Rocío Galán-Megías, María Dolores Lanzarote-Fernández, Javier Casanovas-Lax, Eva María Padilla-Muñoz

**Affiliations:** 1International Doctoral School, University of Seville, 41004 Seville, Spain; rocgalmeg91@gmail.com; 2Department of Personality, Assessment and Psychological Treatments, University of Seville, 41018 Seville, Spain; evapadi@us.es; 3Pediatrics Integral and Pediatric Psychology Research Group CTS-152, Consejería de Salud, Junta de Andalucía, 11620 Andalusia, Spain; 4Paediatrics Clinical Management Unit, University Hospital of Valme, 41014 Sevilla, Spain; javiercasanovas@gmail.com

**Keywords:** adolescents, attention, impulsivity, intelligence, preterm

## Abstract

There is insufficient evidence on the intellectual and attentional profile of adolescents born prematurely. Aim: to identify maladjustment in intellectual and attention skills at the beginning of secondary school. Method: 69 premature 12-year-old adolescents were evaluated with the WISC, d2 Test of Attention, and Test of Perception of Differences-Revised (CARAS-R). Results: adolescents present intellectual and attention abilities in the normal range. However, all premature adolescents show difficulties in impulse control and female adolescents are better in processing speed. Depending on the category of prematurity, differences in attention skills are evident. Conclusion: adolescents born prematurely without associated sequelae have significantly lower performance in the same areas than the normative group. This could affect the cognitive control of their behavior and academic performance in the medium and long term. Great prematurity could interfere with attention skills and self-control even at the age of 12, especially in males.

## 1. Introduction

Worldwide, the rate of preterm births ranges between 5% and 18% of newborns and shows a progressive increase [1]. In Spain, the total number of registered births was 420290 in 2015, of which 23,345 (5.5%) were moderate preterm (32 to 37 weeks of gestational age, GA), 2575 (0.61%) were very preterm (28 to <32 weeks of GA), and 1015 (0.24%) were extremely preterm (<28 weeks of GA) [2].

There are numerous examples of bibliography in which a history of prematurity is related to several behaviors and developmental disorders during childhood. However, there is no consensus among researchers about adolescence. This population arouses special interest since a set of physical, cognitive, affective, and behavioral changes occur in this stage, to which the risks involved in being born preterm are added [3,4].

On this matter, studies referring to adolescents born preterm are limited both worldwide and, in our context, Spain. Furthermore, most of these research studies are based on general and scattered data [5,6,7,8] or parents’ and professors’ reports [8,9,10,11,12].

The relationship between intellectual and attention skills, especially in executive function and other processes involved in behavioral self-regulation, is fundamental to better understand academic performance in the adolescent population without a history of prematurity [13]. Regarding the adolescent population with a history of prematurity, several studies on global intellectual skills found cognitive deficits or lower intelligence quotient (IQ) in premature teenagers [6,14,15], and another concluded that preterm and born-at-term adolescents have similar intellectual performance levels [5,16,17]. However, at this point, it is interesting to study specific intellectual skills and their relationship with attention skills to know if there is a specific cognitive profile derived from the sequelae in this population. Despite the limited bibliography, there is certain evidence with regard to very preterm children. Foulder-Hughes and Cooke [18] found a correlation between performance in intelligence and attention at eight years old. Likewise, Bayless and Stevenson [19] obtained a dependent relationship between executive attention and IQ in a sample of 6- to 13-year-old very premature children.

Summarizing the conclusions of the latest researches, there are scarce, confusing, and inconclusive results related to intellectual and attention skills in early adolescents with a history of prematurity. Moreover, most of the studies have not provided any data about gender differences and between categories of prematurity in adolescents. These gaps in data are relevant in our context since investigations about the preterm population are mainly based on the two first years of life.

The studies that specifically analyze intellectual and attention abilities are needed to help better determine the specific sequelae derived from immaturity; however, the data are still very sparse. Bayless and Stevenson [19] found a relationship between low developments in executive functions and lower IQ in teenagers with a history of prematurity. These adolescents achieved lower scores in all areas of the WISC-R, principally in verbal comprehension and perceptual reasoning tasks. Rose et al. [20] found a lower IQ in 11-year-old preterm children than in their full-term control group on a large battery of 15 tasks. Narberhaus et al. [21], in a sample between 11- and 18-year-old adolescents, summarized that the performance was lower in IQ, verbal, and manipulative index of the WISC-R, although always within the interval of normality. With reference to the existence of differences in the cognitive area related to gender, in general, the previous studies have found a trend of male adolescents having a lower performance [22]. In the research carried out by Linsell et al. [6], the difference is up to eight points in favor of female adolescents, although it was not significant. Other authors such as Cserjesi et al. [23] or Saavalainen et al. [24] also did not obtain any gender-related differences.

Differentiating between two preterm categories in respect to intellectual skills, in a study with 15-year-old late-preterm teenagers, Gurka et al. [25] concluded that their intellectual performance was within the medium range, and there were no differences with those born at term. Nevertheless, extremely preterm teenagers could develop long-term brain abnormalities that could negatively affect their intellectual performance [26]. The same conclusions, with similar samples, were obtained by O’Reilly et al. [7] and Hallin et al. [27], who obtained IQ significantly lower than the control standards. A recent study in our contextconducted by García-Martínez et al. [5], with a sample of 9-to 11-years-old children, focused on perinatal risk, indicating significant differences in IQ and high effect size in perceptual reasoning and memory. Conversely, Grunewaldt et al. [28] found that extremely preterm had the same IQ as those born at term, but, working memory and attention skills were lower in the preterm group.

As regards research studies based on attention skills, one of the first conclusions was that preterm teenagers have a greater risk of developing attention disorders [29], basically in selective attention [10,30] and sustained attention [20]. Other studies corroborate these findings and show that premature teens, similar to premature infants, could develop certain symptoms of attention deficit disorder with or without hyperactivity (ADHD or ADD) [8,10,31,32,33,34]. In a meta-analysis developed by Franz et al. [35], it was shown that preterm teens are three times more likely to be diagnosed with ADHD than those born at term, and Yates et al. [8] concluded that the probability is more than five times. In the same vein, Loe et al. [36] reported a rise and development of attention disorder symptoms in premature children aged between 9 and 16 years old. Contrary to the general population, those in the preterm population reveal no differences associated with gender [11]. Considering the categories, in extremely preterm, Anderson et al. [37] and Wilson-Ching et al. [38] found attention maladjustments, specifically selective and divided attention deficits. Thereby, Yang et al. [39] confirmed the manifestation of these attention disorders in the same risk group, whereby ADHD constitutes one of the main neuropsychiatric and behavioral disorders. Specifically, some studies showed a high difficulty in the control of impulsivity, where sex or prematurity level is not relevant [22,40,41,42].

Nonetheless, in the majority of these studies, the data related to attention and behavior are taken from reports by parents and professors, or self-reports by adolescents, instead of the direct assessment of these skills. Some of the studies in which the children or adolescents were directly evaluated, showed difficulties in impulsivity more than attention deficiency [22,42]. On the other hand, several authors reported that there is a “pure and specific” attention disorder in premature infants that begins in childhood and continues into adolescence [40,43]. The insufficient bibliography related to gender differences provides controversial data [8,11,44].

Therefore, in adolescents born prematurely, there is not a clear profile about intellectual and attention skills neither in gender nor in degree of prematurity. The present study assesses the intellectual and attention skills at the beginning of secondary school in preterm birth adolescents without sequelae who were born before 37 weeks of gestation. The aims of this study are as follows: (1) evaluation of the differences in intellectual skills between adolescents with a history of prematurity and the average scores of the scales of the tests; (2) comparison of the intellectual skills related to the gender and the category of prematurity, i.e., moderate preterm vs. very preterm adolescents; (3) assessment of the possible differences in attention skills between adolescents with a history of prematurity and the average scores of the scales; (4) comparison of the attention abilities regarding the gender and the category of prematurity; (5) analysis of the relationship between intellectual and attention assessments, both in the total sample and when differentiating between categories of prematurity.

## 2. Materials and Method

### 2.1. Participants

The sample was composed of 69 adolescents evaluated at 12 years old, with a history of prematurity, 38 male adolescents (55.1%) and 31 female adolescents (44.9%). All participants had undergone regular specific pediatric and psychological follow-ups. The exclusion criteria were the possession of a different perinatal diagnosis of prematurity or having significant sequelae at the age of 12 years, which would interfere with the assessment process, as sensory, physical, or cognitive significant deficits. The sample was divided into two categories: moderate preterm (≥32 weeks of gestation), composed of 49 adolescents (71%); very preterm (<32 weeks of gestation), with 20 adolescents (29%).

### 2.2. Instruments

The assessment was carried out with three tests—one for intellectual skills and two for attention skills. All of them were applied to the adolescents.

Wechsler Intelligence Scale for Children (WISC-IV) [45] and (WISC-V) [46] provide a complete evaluation of the intellectual ability in children and adolescents, based on the main areas of intelligence: verbal comprehension (VCI), visual–spatial (VSI), fluid reasoning (FRI), working memory (WMI), processing speed (PSI), and total intellectual capacity (IQ). The mean score is 100, and the standard deviation is 15.

D2 Test of Attention [47] assesses several areas of selective attention and concentration. Two measures are used—effectiveness of the test (TOT) and concentration index (CON). 

Test of Perception of Differences-Revised (CARAS-R) [48] evaluates the ability to notice, quickly and correctly, similarities, differences, and stimulating patterns partially organized. Two indices are utilized, attention efficacy (AE) and impulsivity (ICI). The mean score in both tests is 50, and the standard deviation is 20.

All these instruments are standardized and have been used in other studies with premature adolescent samples [17,25,49].

The research protocol was approved by the ethics committee of the university and hospitals (Ethics Code 1292-N-17) where the study was developed. Through a brief telephone interview, and with the same consignment for all the families involved, the aims of this research were explained, and their participation was solicited. On the agreed day, before starting the evaluation, an information sheet about the research was distributed, and written informed consent was signed by all the families involved. The children were individually assessed, in a single session of approximately three hours duration. Following the evaluation, we offered the families the opportunity to receive a return report with results obtained.

### 2.3. Data Analysis

The study is an ex post facto retrospective design. Data analysis was carried out by using IBM SPSS statistics, version 26 for windows. A descriptive and binomial test with a quantitative cut-off point was realized, and Student’s *t*-test or Mann–Whitney U test was performed for analyzing the function of the parametric assumptions. Moreover, Pearson’s and Spearman’s correlations were employed to study the relationship between intellectual and attention skills and prematurity. The effect size with Cohen’s *d* and Rosenthal’s *r* were calculated to quantify the degree of agreement or relationship among the variables studied. The criteria proposed by Cohen [50] were adopted for the interpretation of the effect as follows: up to 0.20, null; 0.20 to 0.50 small; 0.50 to 0.80, medium; more than 0.80, large. The criteria followed by Rosenthal were <0.10 null effect, 0.10 to 0.30 small, 0.30 to 0.50 medium, 0.50 to 0.70 large, and >0.70 very large [51,52].

## 3. Results

Relating to the first aim, i.e., evaluating possible differences in intellectual skills between early adolescents with a history of prematurity and the average scores of the scales of the tests used, results in the binomial test showed that they had an intellectual performance within the normal range. There were significant differences in VCI (*p* < 0.05), FRI, and PSI (*p* < 0.01), and the effect sizes were small. The results highlight that 83% of the preterm adolescents had lower scores than the average in FRI, and 68% in PSI (Table 1).

Regarding the second aim, i.e., the comparison of the intellectual skills related to gender, preterm female adolescents obtained better results than preterm male adolescents in PSI (*p* < 0.008, *r*: −0.316). The distribution was different than expected with respect to the normative average, being significant in FRI, where 79% of the male adolescents and 86% of the female adolescents obtained results below the average (*p* < 0.007, *d*: −0.453; and *p* < 0.001, *r*: −0.255), respectively. Boys obtained low results in PSI by 79% (*p* < 0.000, *d*: −0.606), while only 29% of female adolescents in IQ (*p* < 0.029, *d*: 0.321).

There were no differences between moderate and very preterm in terms of intellectual abilities. Comparing moderate preterm with the normative data, there were significant differences in VCI, FRI, and PSI (Table 2). In moderate preterm, 85% of them achieved lower results in FRI (*p* < 0.000, *d*: 0.514) and 71% in PSI (*p* < 0.004, *d*: 0.370). The very preterm did not show significant differences, compared with the normative data, even though 71% achieved lower outcomes in FRI (see Appendix A).

As regards the third aim, which was the assessment of the possible differences in attention skills between adolescents with a history of prematurity and the average scores of the scales of the tests utilized, after calculating the binomial test, the results indicate performance within the normal range. Nevertheless, there were significant differences in ICI (*p* < 0.00, *r*: −0.371), where the whole sample obtained results below the mean (Table 3).

Concerning the fourth aim, it was compared attention abilities with respect to gender and the category of prematurity. There were no significant differences in gender, but 68% of the male adolescents achieved results below the mean in AE (*p* < 0.034, *r*: −0.04). Additionally, moderate preterm group reached better results than very preterm in TOT (*p* < 0.039, *r*: −0.236) and CON (*p* < 0.024, *r*: −0.268) (Table 4). There were differences in the distribution of the results when the moderate preterm adolescents were compared with the scale in CON, obtaining 65% scores above average (*p* < 0.044, *r*: 0.174) (Table 5). Conversely, 65% of the very preterm had performance below the average in TOT and CON, but the differences were not significant. On the topic of impulsivity, there were no significant differences related to gender or the category of premature.

Finally, regarding the fifth aim of the study, i.e., analysis of the relationship between intellectual and attention assessments, the correlations in the total sample displayed several significant and positive relations. The whole indices of the intellectual skills correlated with the attention indices of the D2 test, TOT, and CON (*p* < 0.01), but less with VCI, which only achieved correlation with CON (*p* < 0.05). The results in the CARAS-R test also showed significant correlations, especially attention efficacy, showing significant correlations with all WISC indices. The impulsivity index is only correlated with VSI (Table 6).

In the moderate preterm category, we found approximately the same significant correlations as in the total premature sample (except VCI with CON, and IQ with TOT) (see Appendix A). In the very preterm category, significant and positive correlations only appear between PSI and IQ with TOT and CON (*p* < 0.01), as well as PSI (*p* < 0.01), VSI, and IQ (*p* < 0.05) with AE (see Appendix A).

## 4. Discussion

The results showed that early adolescents with a history of prematurity and with no associated sequelae have a performance level in intellectual and attention skills that is in the average of the composite scores of the WISC and the attention tests (D2 and CARAS-R) applied. These results situate our sample of teenagers at the population average similar to previous research studies [5,7,21,25], which indicated similar cognitive performance in preterm adolescents. Nevertheless, we found that the preterm early teenagers could show lower levels of competence in tasks that require skills related to fluid reasoning, processing speed, and self-control. According to Taylor et al. [41], these not-so-negative results allow us to inform parents of the high proportion of 12-year-old teenagers born prematurely who have intelligence and attention scores within the average range.

Regarding the intellectual skills, our results agree partially with those that confirmed intellectual performance in preterm infants in the mid-range, although lower than those born at term in perceptual reasoning tasks [19]. These mismatches could lead to specific difficulties in learning or cognitive control of behavior, in this stage of transition to high school. We did not find clear differences with the normative sample; this might be due to the fact that current research does not include adolescents who have any sequelae, as other anterior studies [5]. Additionally, the whole sample of teenagers was included in the following program until the age of 6 years old. However, this fact does not assure that the participants in those types of follow-up programs are also involved in an intervention process. Conversely, our results contrast with another study that indicated lower language proficiency in preterm adolescents than those born at term [19].

Evidence reveals some gender-related differences in the early teenagers who were born preterm; females adolescents achieved better performance in processing speed, higher than the percentage of males who could find difficulties in this competence. Both gender groups could have difficulties in tasks associated with fluid reasoning. Finally, preterm female adolescents achieved better performance in intellectual capacity skills than the general population. Our results do not support that the male adolescents have lower achievements in all levels of intellectual competence [6,22], although we cannot affirm that there are no differences either [23].

When we differentiate between preterm categories, the results only indicate differences between the group of moderate preterm teenagers and the general population. Moderate preterm teenagers showed more difficulties in tasks related to fluid reasoning and processing speed, for which a high percentage performed below the average. The very preterm group obtained no significant results, even though a great percentage of them achieved low results in fluid reasoning. These data do not support previous investigations in which no differences were obtained when comparing moderate preterm teenagers with those born at term [25], nor do they support those studies that show differences in the very premature category [7,26,27].

However, we found that adolescents in either category obtained a normal level of performance in intellectual abilities, and there are no differences between premature categories. These results are partially consistent with previous ones that reported similar performance on all the subscales of the WISC in both categories of preterm adolescents, except for working memory [28].

On the subject of attention skills, our teenager group achieved scores similar to the population average, except in impulsivity. Our results do not maintain the idea of the existence of attention deficits in this population, a higher risk of developing attention maladjustments, or symptoms of ADD or ADHD [8,10,20,29,30,31,32,33,34,35,36].

In spite of this, our results agree with some authors [22,42] who directly assessed adolescents or through meta-analyses [40]. All these investigators concluded that preterm teenagers behave more impulsively than those born at term, and they warn that this could be related to problems in control and regulation of behavior. In this line, parents and teachers reported more difficulties in the behavioral areas assessed, including attention, impulsivity, hyperactivity, working memory, and planning–organization [10]. Relevant to this fact, the complete sample, regardless of the prematurity category or gender, obtains results that indicate difficulties for self-control.

Concerning gender and attention skills, there were no differences between female and male adolescents in a sample that evaluated those 11 years old [11]. Our findings agree with those studies in which male achieve lower than female adolescents [8,10,37,44]. When comparing the scale, the whole sample of preterm adolescents show low self-control, and male adolescents have more difficulties in attention efficacy. These data do not corroborate the absence of differences in attention between preterm and born at term [10,37].

As regards the differences between categories of preterm, we found that the very preterm attains an average performance, although it is significantly lower than moderate in tasks that require selective attention and concentration. These results are similar to those that pointed to the very preterm category as the most vulnerable population to develop attention deficits [37,38,39]. Moderate premature teenagers, unlike very preterm, perform better than the general population in concentration levels. These data could support the idea of the atypical profile in premature infants [53].

Together, the results found do not support the hypothesis that adolescents born prematurely have symptoms consistent with ADD or ADHD. Achievements in attention are adequate but not in self-control. More attention difficulties are evident in adolescents born before the 32nd week of pregnancy. In this research, the teenagers were directly evaluated, through attention tests or intelligence subtests and not through self-reports or questionnaires completed by parents or teachers, as in other investigations [8,10,11]. High impulsivity might confuse caregivers about their attention abilities and therefore needs further investigation.

Finally, in terms of the association between intellectual and attention skills in premature, our findings are corroborated [18,19]. In this respect, the relationship between these two skills in 12-year-old early adolescents born prematurely, in general, is supported, but not as much in very preterm. More specifically, our results indicate that intellectual abilities are directly and positively related to the evaluated attention indices; that is, it could be confirmed that there is a relationship between the intellectual skills evaluated with the WISC and the ability to focus on one or two important stimuli while deliberately suppressing the awareness of other distracting stimuli, thereby maintaining concentration on the task at hand. We also observed a positive relationship between the actual effectiveness of completing a given task and the ability to process and integrate spatially information, organize it, and explore it quickly. These abilities and the brain systems involved develop throughout childhood and are frequently compromised in premature infants, even those with an average global cognitive ability [41], as has been the case in the present study. All of these abilities are related to academic achievement. Therefore, these findings indicate that academic achievement or failure cannot be correctly interpreted without taking into account the characteristics of the attention, and this is in the same line as that proposed for adolescents in general [13]. Analyzing the relationships between these two skills according to the category of prematurity, very similar relations are found in the moderate preterm category. Conversely, in the very preterm category, less positive and direct correlations are obtained. This might indicate the existence of other uncontrolled variables that could be playing a mediating role in the results. Therefore, the ability to focus on important stimuli, to maintain concentration during the execution of the task, and efficacy in doing the task, is related to the capacity for fluid reasoning index and the total intellectual capacity in the very preterm category. Attention is a basic ability for the learning process. The poor correlations with intelligence outcomes in very preterm, in addition to the heterogeneity of their performance, make long-term follow-up more necessary. It is important to note that the small size of the sample of very premature presents a great limitation for this research, and therefore, although it is a population with a lower incidence, it would be noteworthy to confirm these results in a large sample. For future studies, with a large population and a more homogeneous nature for the different categories of prematurity, the hypothesis that moderate and very preterm teenagers have different intellectual and attention profiles could be advanced.

The current study collected the outcome of teenagers experiencing a period of educational transition (primary to secondary school), in an environment where the follow-up is done rarely; Being this fact the reason why previous studies related to the population at risk in this age range are limited.

Thus, our findings allow us to advance the idea that early adolescents born preterm without associated sequelae show adequate cognitive skills, although with a higher risk of presenting difficulties in reasoning and information processing, which are also influenced by the category of prematurity and gender. Furthermore, along the lines indicated by Linsell et al. [6], the data clarify the confusion about whether there is a higher prevalence of general attention deficits, ADD, or ADHD in those born prematurely, or whether it is a case of a deficit of attention related to the level of intellectual competence, as certain authors have indicated [40,43]. Our results seem to be heading in another direction [42]. The hypothesis could be that self-control difficulties might be associated with misdiagnoses of attention deficits based on information from parents and teachers [8,10,11,35]. For this reason, we propose to incorporate information from parents and adolescents in future works.

In conclusion, early adolescents with a history of prematurity and without associated sequelae could have a performance level within the population average. However, in some cognitive abilities, they might have higher difficulties, fundamentally in self-control. Male adolescents could have more problems in processing speed, and those born very prematurely show more difficulty in attention. These findings allow us to advance the hypotheses of atypicality in the cognitive profiles of premature infants [53] and the delay in the acquisition of skills rather than the hypotheses referring to the irreversible deficit of these skills.

Regarding the possible implications of our study for future investigations, the results obtained lead us to propose new lines of research. On the one hand, it would be of interest to conduct a mediation analysis in which the attentional variables, the weeks of gestation, and some cognitive variables were involved. On the other hand, it would be very convenient to address impulsivity and inhibitory control in adolescents born prematurely, regardless of gender and prematurity category. It is also relevant to investigate the competencies in attention skills evaluated directly in samples of premature adolescents in addition to evaluations in their parents, as well as the possible interaction between these two measurement sources. 

Furthermore, based on these results, it would be interesting to conduct a more specific evaluation of the self-control of premature adolescents. This would reduce the impact on the incidence of impulse control disorders in later stages of life. Additionally, due to the absence of clear cognitive deficits, if efforts are made to increase self-control, their academic achievement will most likely progress in the right direction. Moreover, the associations found between the skills studied and the suitable environments suggest how early interventions and the incorporation of specific educational programs could mitigate these risks [13,41]. These interventions should be focused on the most vulnerable group—children born very prematurely. To this end, in our context, a protocol should be established to evaluate all premature children in transition to secondary school because developmental tests administered in the first years of life are not good predictors of performance levels at later ages [41]. This follow-up should include a comprehensive neurodevelopmental assessment to prevent difficulties and promote academic and personal achievement. 

## Figures and Tables

**Table 1 ijerph-18-09043-t001:** Comparison of the average of the composite scores of intellectual skills in adolescents born prematurely and normative group (scales).

	Preterm Adolescents		Binomial Test: Mean of the Normative Group		
Intellectual Skills	Mean	SD	g.1.%	*p*	Effect
Verbal Comprehension (VCI)	106.75	15.216	35	**0.015**	0.446 ^a^
Visual–Spatial (VSI)	99.07	17.685	52	0.810	0.056 ^a^
Fluid Reasoning (FRI)	93.04	13.568	83	**0.000**	0.486 ^a^
Working Memory (WMI)	100.84	16.034	55	0.470	0.054 ^a^
**Processing Speed (PSI)**	94.88	12.676	68	**0.004**	0.368 ^a^
Intellectual Capacity (IQ)	100.32	18.470	45	0.470	0.009 ^b^

Note: SD = standard deviation; g.1. = % below the mean of the group of premature adolescents; ^a^ = Cohen’s *d*; ^b^ = Rosenthal’s *r*. Significant data in bold.

**Table 2 ijerph-18-09043-t002:** Comparison of the average of the composite scores of intellectual skills in moderate and very preterm adolescents and the normative group (scales).

Binominal Test	Moderate Premature			Very Premature		
Intellectual Skills	g.1.%	*p*	Effect	g.1.%	*p*	Effect
Verbal Comprehension (VCI)	35	**0.044**	0.200 ^a^	35	0.263	0.556 ^b^
Visual–Spatial (VS)	51	1	−0.028 ^a^	55	0.824	0.053 ^b^
Fluid Reasoning (FRI)	85	**0.000**	0.514 ^b^	71	0.453	0.306 ^b^
Working Memory (WMI)	53	0.775	0.033 ^b^	60	0.503	0.103 ^b^
Processing Speed (PSI)	71	**0.004**	0.370 ^b^	60	0.503	0.359 ^b^
Intellectual Capacity (IQ)	45	0.568	0.002 ^a^	45	0.824	0.068 ^b^

Note: SD = standard deviation; g.1. = % below the mean of the group of premature adolescents; ^a^ = Rosenthal’s *r*; ^b^ = Cohen’s *d*. Significant data in bold.

**Table 3 ijerph-18-09043-t003:** Comparison of the average percentiles index of attentional skills in adolescents born prematurely and the average percentiles of D2 and CARAS-R tests.

	Premature Adolescents		Binomial Test: Mean of the Normative Group		
Attention Skills	Mean	SD	g.1.%	*p*	Effect
Selective Attention (TOT)	52.74	26.232	49	1	0.058 ^a^
Concentration Level (CON)	53.78	26.594	43	0.336	0.160 ^b^
Attentional Efficacy (AE)	50.97	25.358	57	0.336	0.042 ^b^
Impulsivity (ICI)	36.09	14.320	100	**0.000**	−0.371 ^a^

Note: SD = standard deviation; g.1. = % below the mean of the group of premature adolescents; ^a^ = Rosenthal’s *r*; ^b^ = Cohen’s *d*. Significant data in bold.

**Table 4 ijerph-18-09043-t004:** Average reached in the attention skills according to the preterm categories.

D2 and CARAS-R	Preterm Categories	M	SD	Sig.	Effect
Selective Attention (TOT)	Moderate Preterm Very Preterm	56.47 43.60	24.766 28.096	**0.039 ^a^**	−0.236 ^c^
Concentration Level (CON)	Moderate Preterm Very Preterm	58.06 43.30	25.138 27.781	**0.024 ^a^**	−0.268 ^c^
Attentional Efficacy (AE)	Moderate Preterm Very Preterm	49.71 54.05	25.974 24.145	0.523 ^b^	−0.173 ^d^
Impulsivity Control Index (ICI)	Moderate Preterm Very Preterm	34.69 39.50	14.628 13.268	0.162 ^a^	−0.169 ^c^

Note: ^a^ = Mann–Whitney U test; ^b^ = Student’s *t*-test; ^c^ = Rosenthal’s *r*; ^d^ = Cohen’s *d*. Significant data in bold.

**Table 5 ijerph-18-09043-t005:** Comparison of the average percentiles index of attentional skills in adolescents born moderate and very premature and the average percentiles of D2 and CARAS-R tests.

	Binomial Test: Mean of the Normative Group					
	Moderate Premature			Very Premature		
Attention Skills	g.1.%	*p*	Effect	g.1.%	*p*	Effect
Selective Attention (TOT)	43	0.392	0.281 ^a^	65	0.263	0.262 ^a^
Concentration Level (CON)	35	**0.044**	0.174 ^b^	65	0.263	0.276 ^a^
Attentional Efficacy (AE)	59	0.253	0.012 ^a^	50	1	0.182 ^a^
Impulsivity (ICI)	100	**0.000**	−0.400 ^b^	100	**0.000**	−0.295 ^b^

Note. SD = standard deviation; g.1. = % below the mean of the group of premature adolescents; ^a^ = Cohen’s *d*; ^b^ = Rosenthal’s *r*. Significant data in bold.

**Table 6 ijerph-18-09043-t006:** Correlations between intellectual and attention skills in the total sample of preterm adolescents.

	VCI	VSI	FRI	WMI	PSI	IQ
TOT	0.214 ^a^	**0.320 ^a^**	**0.387 ^a^**	**0.328 ^a^**	**0.422 ^a^**	**0.404 ^a^**
Sig.	0.077	0.007	0.008	0.006	0.000	0.001
CON	**0.259 ^b^**	**0.392 ^b^**	**0.504 ^b^**	**0.329 ^b^**	**0.481 ^b^**	**0.445 ^a^**
Sig.	0.031	0.001	0.000	0.006	0.000	0.000
AE	**0.380 ^b^**	**0.324 ^b^**	**0.510 ^b^**	**0.277 ^b^**	**0.548 ^b^**	**0.491 ^a^**
Sig.	0.001	0.007	0.000	0.021	0.000	0.000
ICI	0.098 ^a^	**0.316 ^a^**	0.137 ^a^	0.085 ^a^	0.098 ^a^	0.213 ^a^
Sig.	0.424	0.008	0.366	0.486	0.423	0.079

Note: TOT = selective attention; CON = concentration index; AE = attentional efficacy; ICI = impulsivity control index; VCI = verbal comprehension index; VSI = visual–spatial index; FRI = fluid reasoning index; WMI = working memory index; PSI = processing speed index; IQ = total intellectual capacity. ^a^ = Spearman correlation; ^b^ = Pearson correlation. Significant data in bold.

## Data Availability

The data presented in this study are available on request from the corresponding author.

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
