# Peer review of "Interaction of Impulsivity, Attention, and Intelligence in Early Adolescents Born Preterm without Sequelae"

_ijerph, 2021, doi:10.3390/ijerph18179043_

Round 1

Reviewer 1 Report

The results of the a priori study seem to be consistent with the objectives set out in the article, although it is no less true that it is described in an excessively brief and superficial way. Despite this, they conform to the methodological approaches established and planned in the study and are, for the most part, conveniently organized and written, at least from a chronological and statistical point of view. 

The conclusions of the study are correctly stated, organized and described. However, despite this, they are unable to clearly delimit the scientific space through which the different empirical studies that, in the future, intend to continue with the trail or the path that they have stopped exploring will have to continue or run the present study. 

The references are quite current, despite the fact that in some cases they end up being older than ten years, and are reflected in the text of the article, scrupulously respecting the APA regulations in its seventh edition.

Author Response

Thank you very much for your comments and contributions.

In the writing of the results, brevity was prioritised due to the large volume of information, with the aim of making them easier to read.

Regarding the line of continuity of this research, some new information has been added in the last paragraph of the discussion. Currently, the authors` aim is to find out the attention competencies of adolescents born prematurely, taking into account the parents' assessment. Also, the possible interaction between parental information and the results provided by the adolescent themselves. This will be a new article in which we are working on, and we hope to publish soon. In addition, based on the results obtained, we believe that a more specific evaluation of the self-control of preterm adolescents would be quite interesting. 

Finally, in response to your request, an English revision has been made.

Reviewer 2 Report

First of all, I would like to congratulate the authors of this paper. The topic they address is relevant, interesting and transferable to the field of early care and educational orientation. I agree with their assertion that it is essential to be able to provide parents of premature babies with a positive, realistic and evidence-based message about the possible challenges that may be faced in the academic environment. For instance, I find this result very relevant: "achievements in attention are adequate, but not in self-control. More attention difficulties are evident in adolescents born before the 32nd week of pregnancy" or "due to the absence of clear cognitive deficits, if we work to increase self-control, their academic achievement will be most likely to progress in the right direction". These are very valuable and transferable contributions, which justify the publication of this study.

Among the contributions, I would like to highlight the use of a more demanding and rigorous evaluation methodology than previous studies, since in this research, where teenagers have been directly evaluated, through attention tests or intelligence subtests, and not through self-reports or questionnaires completed by parents or teachers, as in other investigations.

The treatment of the data obtained is presented in detail and the statistical tests used are justified. For example, a distinction is made between variables that are normally distributed and those that are not and test was performed in function of the parametric assumptions, alpha values are accompanied by tests of effect size, etc.

Participants:

It is clear that the number of participants is reduced, as is often the situation in studies working with special populations, which have to meet strict selection requirements. However, the inclusion and exclusion criteria, which in the corresponding section are reduced to the expression adolescents born prematurely "without associated sequelae", should be further developed. Since this criterion is a very important element in the study, it should be made clearer. In the case of the age, the authors perfectly justify the chosen age.

Design and Procedure: The study design is not well described. In the Data Analysis section, where the authors note The study is an "ex-post factor", it should read "ex-post facto". Nor do I agree with the term "prospective design". In the explanation of the procedure for obtaining the data, the authors state textually. "The children were individually assessed, in a single session of approximately three hours' duration". When data are collected at a single point in time, we speak of ex-post facto retrospective designs. The authors should review this designation.

Results: Although, as we have mentioned, care has been taken in the treatment of the data, it is lacking that the authors go deeper into the relationships between the variables studied and provide some analysis that seeks a joint relationship between them. The data provided seem to indicate that attentional variables may have a mediating or moderating effect between the week of gestation and some variables assessed by cognitive tests such as processing speed. It is clear that the number of participants is very small, but we are sure that this effort will lead to very interesting results.

Discussion: In this section they break with the standard type of citation and accompany the reference numbers of the citations with the surnames of the authors. It is understood that this resource can be used, at some specific moment, when authors are used as subjects of the sentence, but this should be the exception and not the norm. Authors should review the citation format of this section of the article.

Finally, it is suggested that more information is provided on the therapeutic attention received by the participants. This could have some influence on the results obtained. It would be very interesting for the scientific community to know the possible early stimulation treatments that could justify these differences or, at least, to offer some information about them in the section describing the participants. On the other hand, if one of the objectives of the study is to provide information from parents and teachers on the characteristics of preterm births, it also seems very important to us to provide the advantages of the interventions developed with this group, in fact, at the end of the discussion it seems that only children born very prematurely receive early intervention. We would like to have more information on this aspect.

Author Response

Thank you very much for your contributions, comments and the positive assessment of this research.

In response to your comments, the concept of “without associated sequelae" has been clarified and the type of design has been modified.

We appreciate the suggestion to carry out a mediation study between attentional variables, weeks of gestation and some cognitive variables. We believe that it would provide interesting information and it has been pointed out in the discussion. Nevertheless, due to the volume of results presented in this article, we have considered this analysis for a new article.

The discussion section has been revised to avoid referencing authors.

Providing relevant information to parents and teachers as consequence of the findings is an implicit aim in any research with risk samples, as practical implications of this work.

In our context, children born prematurely below 32 weeks of gestation are monitored up to the age of 6 years, and they are treated with Early Intervention when any difficulty or developmental delay is detected. This supervision is centred on the child, providing guidance to parents and/or teachers. The guidelines in early intervention are very diverse, and depend on the characteristics of the child and the resources available in each centre. Thus, this data has not been included in the evaluation.